cognition/psychology

theory of mind, Eyes Test, emotional intelligence, MSCEIT

**Author for correspondence:**
María José Gutiérrez-Cobo
e-mail: mjgc@uma.es

# The 'Reading the mind in the Eyes' test and emotional intelligence

Alberto Megías-Robles[1], María José Gutiérrez-Cobo[2],
Rosario Cabello[3], Raquel Gómez-Leal[1],
Simon Baron-Cohen[4] and Pablo Fernández-Berrocal[1]

[1]Department of Basic Psychology, Faculty of Psychology, University of Málaga, Spain
[2]Department of Developmental and Educational Psychology, Faculty of Psychology, University of Málaga, Campus Teatinos, s/n. 29071 Spain
[3]Department of Developmental and Educational Psychology, Faculty of Psychology, University of Granada, Spain
[4]Autism Research Centre, Department of Psychiatry, University of Cambridge, Cambridge, UK

  AM-R, 0000-0002-4563-7465; MJG-C, 0000-0002-8219-5133;
RC, 0000-0001-6908-7181; RG-L, 0000-0003-1847-6966;
SB-C, 0000-0001-9217-2544; PF-B, 0000-0002-0844-2976

The 'Reading the Mind in the Eyes' test (Eyes Test) has been widely used to measure theory of mind (ToM) or the ability to recognize the thoughts and feelings of others. Although previous studies have analysed its relationship with the ability to perceive emotions, the potential links with more complex emotional abilities remain unclear. The aim of the present research was to analyse the relationship between the Eyes Test and each of the emotional intelligence (EI) branches: perceiving, facilitating, understanding and managing emotions. In addition, we were interested in studying these relationships as a function of the Eyes Test difficulty. Eight hundred and seventy-four participants completed the Eyes Test and the Mayer–Salovey–Caruso Emotional Intelligence Test. A stepwise multiple regression analysis for the total score on the Eyes Test revealed that the best fitting model included the understanding, perceiving and managing emotion branches, with the understanding branch being the one most strongly associated with performance on the Eyes Test. Interestingly, stepwise multiple regression analysis for the easiest items of the Eyes Test revealed the same predictors, but, in the case of the most difficult items only the understanding branch was a predictor. These outcomes were not moderated by the influence of gender. Our findings support the notion that the Eyes Test can be used as a ToM task and that it is associated with complex EI abilities. Limitations and future lines of investigation are discussed.

# 1. Introduction

Theory of mind (ToM) is the ability to recognize the thinking or feelings of others in order to predict their behaviours and act accordingly [1]. Deficits in ToM have been found in a variety of clinical conditions such as autism, social anxiety, depression or bipolar disorder [2–4]. There is a variety of ToM measures employing different approaches. For instance, some instruments are supposed to measure explicit verbal reasoning, such as the Strange Stories task [5] or the Faux Pas task [6], while others entail a more implicit social analysis, an example of the latter being the 'Reading the Mind in the Eyes' test (Eyes Test) [7,8], a widely used instrument to measure ToM in adults. This is a 36 item test where participants have to indicate which emotion best matches the mental state that different eye images display.

Emotional intelligence (EI) is an ability composed of four branches of increasing complexity: perceiving, facilitating, understanding and managing emotions. It is defined as 'the ability to perceive accurately, appraise, and express emotion; the ability to access and/or generate feelings when they facilitate thought; the ability to understand emotion and emotional knowledge; and the ability to regulate emotions to promote emotional and intellectual growth' [9, p. 10]. The Mayer–Salovey–Caruso Emotional Intelligence Test (MSCEIT, [10]) is the main instrument used for assessing this hierarchical model. This instrument measures EI in an objective manner through the resolution of emotional problems with correct and incorrect answers. Although MSCEIT has shown good psychometric properties and is the most well-established tool for assessing EI ability through performance measures, it is important to note that it is not exempt from certain limitations [11–13]. For instance, Fiori et al. [12] observed that this instrument could be more suitable for those participants with deficiencies in EI, as it is less able to distinguish between individuals with high EI scores. Olderback et al. [11] have also proposed that the perceiving emotions branch of the MSCEIT appears to operate differently to other emotion perception ability tests.

Following previous suggestions on the basis of research employing the Eyes Test [14–16], relating EI with the Eyes Test could help to better understand the characteristics of this latter instrument and confirm whether it is associated with more complex emotional abilities.

The current literature already includes some studies that have analysed the relationship between the performance on the Eyes Test and emotional abilities. For example, several studies have found moderate correlations between the Eyes Test and the ability to perceive emotions, also when using event-related potential [16–18]. Other investigations have focused on assessing the relationship between performance on the Eyes Test and specific EI branches [19,20]. For instance, Maillefer et al. [19] observed a significant positive correlation between the understanding branch of EI and Eyes Test scores while Ferguson & Austin [20] found no significant results for either the understanding or the managing branches. In addition to these results, Warrier et al. [15] proposed that the Eyes Test could show a verbal component typical of the understanding of emotion ability, which is worthy of further exploration.

Given these mixed results, the aim of the present study was to analyse more in-depth the relationship between the Eyes Test and EI in order to identify which EI branches are most strongly associated with global performance on the Eyes Test. In addition, we were interested in studying this relationship as a function of the complexity of the items of the Eyes Test (easier versus more difficult). The aim of separately analysing these items according to their level of difficulty was to explore whether the most difficult items involve the use of more complex emotional abilities. The results of this study would allow us to more deeply understand the association between the Eyes Test and emotional abilities that are rather more complex than the perception of emotions, such as the understanding or managing emotion ability. Finally, given that previous studies have found gender differences in EI and Eyes Test performance in favour of females [21,22], we also explored if gender acts as a moderating factor in the relationship between these two constructs.

In summary, the present study addresses these issues in two novel ways. First, we wanted to include the MSCEIT as a measuring instrument, which encompasses the whole model of EI [9], thus allowing for an in-depth understanding of the relationships between Eyes Test scores and the EI branches. Second, our study offers the potential to further analyse the relationship between the two variables by examining the Eyes Test items according to their level of difficulty.

# 2. Materials and methods

## 2.1. Participants

Eight hundred and seventy-four participants from different universities and Spanish community samples voluntarily agreed to take part in this study. The sample was recruited by advertisements in universities,

social networks and online platforms. As compensation for their involvement, the participants were offered a report describing their emotional intelligence abilities. The sample comprised 182 men and 692 women, with a mean age of 22.44 years (s.d. = 4.52: age range: 18 to 60). The Research Ethics Committee of the University of Málaga approved the study protocol (14-2019-H) as part of the project PSI2017-84170-R. Participants gave informed consent and were assessed in accordance with the Helsinki declaration [23].

## 2.2. Materials

### 2.2.1. The 'Reading the Mind in the Eyes' test [8]

This test includes 36 photographs of male and female eyes depicting emotional states. For each photograph, participants are asked to choose the emotional state that best describes the eyes, choosing between one of four possible emotions. In the present study, the performance of the participants was calculated as the number of correct responses divided by the total number of trials (36 photographs). We used the Spanish version of the Eyes Test [7]. The internal consistency for the sample of our study was acceptable (ordinal Cronbach's $\alpha = 0.67$).

### 2.2.2. The Mayer–Salovey–Caruso Emotional Intelligence Test [10]

This instrument is a performance-based ability measure of EI composed of 141 items divided into four branches according to Mayer and Salovey's theory: perceiving, facilitating, understanding and managing emotions [9]. In our study, EI abilities were measured using the Spanish version, which has shown good internal consistency (Cronbach's $\alpha = 0.95$; [24]). In our sample, the internal consistency for the MSCEIT total score was good (Cronbach's $\alpha = 0.86$) and for the MSCEIT branches this ranged between questionable and good (Cronbach's $\alpha$ for MSCEIT perceiving = 0.84; Cronbach's $\alpha$ for MSCEIT facilitating = 0.63; Cronbach's $\alpha$ for MSCEIT understanding = 0.61; and Cronbach's $\alpha$ for MSCEIT managing = 0.77).

## 2.3. Statistical analysis

First, descriptive statistics, along with gender differences, were computed for the Eyes Test performance and MSCEIT scores. Second, Pearson's correlations were conducted to test for the existence of significant relationships between each of the variables included in the study. Third, in order to explore the effect of gender in more depth, we conducted a series of moderation analyses with gender as the moderating factor of the relationship between MSCEIT scores and Eyes Test performance. The variables were mean-centred. Fourth, we conducted stepwise regression analysis to identify the set of MSCEIT branches that best explain performance on the Eyes Test. In addition, to address the objectives of the study, we also decided to calculate an index of Eyes Test performance for 50% of the easiest items (Eyes Test easy items) and another index with 50% of the most difficult items (Eyes Test difficult items). In order to avoid circular analysis, the items included in each index were selected according to the original study by Fernández-Abascal *et al.* [7].[1], which validated the Spanish version of the questionnaire. A *t*-test confirmed the existence of significant differences between the groups of easy and difficult items in our sample ($t_{873} = 27.74$, $p < 0.0001$, Cohen's $d = 0.94$). Pearson's correlations, moderation analyses and stepwise regressions were also conducted for both indices. The analyses were carried out using the SPSS 24 software (IBM corp., USA) and SPSS PROCESS macro 3.4 [25].

## 3. Results

Descriptive statistics of the variables included in the study are shown in table 1. The analysis of the gender differences revealed that women on average obtained significantly higher scores than men on MSCEIT total, MSCEIT facilitating, MSCEIT managing, Eyes Test total and Eyes Test easy items. Pearson's correlations revealed that both MSCEIT total and the four branches of the MSCEIT were positively correlated with the performance on the Eyes Test task (both on the total score, and easy and difficult items; all $p < 0.05$; table 2). Moderation analyses did not reveal any significant effect of gender as a moderator of the

---

[1]Items included in the category of 'Eyes Test easy items': 4, 5, 8, 9, 12, 13, 14, 15, 16, 18, 20, 21, 28, 29, 30, 32, 35, 36. Items included in the category of 'Eyes Test difficult items': 1, 2, 3, 6, 7, 10, 11, 17, 19, 22, 23, 24, 25, 26, 27, 31, 33, 34.

**Table 1.** Descriptive statistics (mean and standard deviation (s.d.)) for the global sample and sample split by gender, and *t*-tests (*t*-value and Cohen's *d*) comparing genders for MSCEIT and Eyes Test scores.

| | global sample | | men | | women | | | |
|---|---|---|---|---|---|---|---|---|
| | mean | s.d. | mean | s.d. | mean | s.d. | *t* | Coheńs *d* |
| MSCEIT total | 108.45 | 7.78 | 106.64 | 8.22 | 108.93 | 7.59 | 3.56** | 0.28 |
| MSCEIT perceiving | 106.15 | 10.25 | 105.54 | 10.72 | 106.31 | 12.12 | 0.90 | 0.07 |
| MSCEIT facilitating | 103.37 | 9.04 | 101.50 | 9.13 | 103.86 | 8.96 | 3.16** | 0.26 |
| MSCEIT understanding | 109.44 | 7.94 | 108.96 | 8.58 | 109.57 | 7.76 | 0.92 | 0.07 |
| MSCEIT managing | 108.34 | 11.12 | 104.82 | 12.27 | 109.27 | 10.61 | 4.87** | 0.39 |
| Eyes Test total | 0.74 | 0.09 | 0.73 | 0.09 | 0.74 | 0.09 | 2.24* | 0.11 |
| Eyes Test easy items | 0.81 | 0.11 | 0.78 | 0.10 | 0.81 | 0.11 | 4.20** | 0.29 |
| Eyes Test difficult items | 0.67 | 0.12 | 0.68 | 0.12 | 0.67 | 0.12 | 0.38 | 0.08 |

**Table 2.** Pearson's correlations between MSCEIT and Eyes Test scores. *p < 0.05; **p < 0.01.

| | MSCEIT perceiving | MSCEIT facilitating | MSCEIT underst. | MSCEIT managing | Eyes Test total | Eyes Test easy items | Eyes Test difficult items |
|---|---|---|---|---|---|---|---|
| MSCEIT total | 0.76** | 0.68** | 0.54** | 0.58** | 0.29** | 0.29** | 0.17** |
| MSCEIT perceiving | | 0.39** | 0.17** | 0.16** | 0.16** | 0.17** | 0.08* |
| MSCEIT facilitating | | | 0.19** | 0.24** | 0.15** | 0.16** | 0.08* |
| MSCEIT understanding | | | | 0.19** | 0.32** | 0.29** | 0.21** |
| MSCEIT managing | | | | | 0.15** | 0.14** | 0.10** |
| Eyes Test total | | | | | | 0.75** | 0.81** |
| Eyes Test easy items | | | | | | | 0.22** |

relationship between MSCEIT (total and branches) and Eyes Test performance (total, easy and difficult items; all $p > 0.05$; estimates for each model are reported in the electronic supplementary material, table S1).

With respect to the regression analysis, first, we observed that the multicollinearity assumption was met for all predictors (all variance inflation factor ≤ 1.23 and tolerance ≥ 0.95). The stepwise multiple regression analysis with Eyes Test total performance as the dependent variable revealed that the best fitting model (11.8% of explained variance) included the understanding ($\beta = 0.28$, $t_{870} = 8.71$, $p < 0.001$), perceiving ($\beta = 0.09$, $t_{870} = 2.91$, $p < 0.01$), and managing ($\beta = 0.08$, $t_{870} = 2.50$, $p < 0.05$) branches. With respect to the distinction between the easy and difficult items of the Eyes Test, the stepwise regression analysis for the Eyes Test easy items identified a model (10.6% explained variance) that included the understanding ($\beta = 0.26$, $t_{870} = 7.88$, $p < 0.001$), perceiving ($\beta = 0.12$, $t_{870} = 3.60$, $p < 0.001$) and managing ($\beta = 0.07$, $t_{870} = 2.13$, $p < 0.05$) branches. The regression analysis for the Eyes Test difficult items only included the understanding branch ($\beta = 0.21$, $t_{872} = 6.31$, $p < 0.001$; 4.4% of explained variance). The full results of the stepwise multiple regressions are shown in the electronic supplementary material, tables S2–S4.

## 4. Discussion

The present study aimed to take a step forward in understanding the emotional processes underlying performance on the Eyes Test by analysing its relationships with the EI branches. In particular, we explored the possibility that the Eyes Test is not only associated with the perception of emotions ability, but also with more complex abilities such as the understanding and managing emotion abilities. Further, we explored the relationship between Eyes Test performance and the EI branches

according to the level of complexity of the former by separately analysing the scores obtained on the easiest and most difficult items. Finally, we explored whether gender was a moderating factor in the relationship between EI and performance on the Eyes Test.

Results for the Eyes Test total performance revealed three EI branches to be included in the best fitting model: understanding, perceiving and managing emotions. Consistent with previous research [16–18], the perceiving emotion branch was linked to better performance on the task. More importantly for this line of investigation, our study found the understanding emotion branch to have the strongest relationship with overall performance on the Eyes Test.

Additional analyses were also conducted to elucidate the EI branches associated with the easiest and most difficult Eyes Test items. In this regard, these stepwise regression analyses revealed the most interesting results. While for the easiest items, the same three previous branches were related to performance on the Eyes Test (understanding, perceiving, and managing); for the complex items, only the understanding branch was related to Eyes Test performance.

Taken together, these results reveal that the Eyes Test can be more than an emotion recognition test [16–18]. In fact, the understanding emotion branch was the variable most strongly associated with performance on the Eyes Test. Interestingly, when items are easy to solve, the emotional perception ability takes part in the process, together with the ability to understand and manage emotions; however, when the task becomes more difficult, it seems that a greater recruitment of more complex emotional abilities, such as understanding emotions, is needed to determine the correct answer. This is consistent with the suggestion put forward by Warrier *et al.* [15], that the Eyes Test is composed of a verbal component that includes a mental state lexicon. This mental lexicon is part of the understanding branch definition which includes the emotional vocabulary of the individual [26]. Although these results contribute towards a better understanding of the processes underlying the performance on the Eyes Test, it is important to consider that the MSCEIT branches included in the regression models account for only a small percentage of the variance of this performance. Thus, additional individual differences could also explain the Eyes Test scores. Given the results of previous studies linking MSCEIT and the Eyes Test scores with fluid and crystallized intelligence [11,14,16,27], future research should include cognitive intelligence measures in order to achieve a more complete integration of the processes underlying the performance on this test.

In addition to these results, gender differences in the target variables were also analysed. Consistent with previous findings [21,22], women scored higher on the EI and Eyes Test than men. Given this result, we analysed whether there were gender differences in terms of the relationship between performance on the Eyes Test and EI. These analyses revealed that gender did not moderate the relationship between the branches of the MSCEIT and the Eyes Test (for either the total or the easiest/most difficult items). However, it is important to note that one limitation of this study is the gender imbalance in our sample, with 79% of the participants being women. Finally, in order to address some of the limitations associated with the MSCEIT [11–13] and Eyes Test, in future research it would be of interest to attempt to replicate the present findings using additional instruments. For example, the short-form solution of the Eyes Test proposed by Olderbak *et al.* [16] has been shown to have higher internal consistency.

## 5. Conclusion

In conclusion, it appears that the Eyes Test could be used as a ToM measure for assessing the skills of understanding mental states. Although these results help to shed light on the debate regarding the nature of the Eyes Test and its relationship with EI abilities [14], future studies should be conducted to reinforce these conclusions by using causal methodologies and additional EI measuring instruments.

Ethics. The Research Ethics Committee of the University of Málaga approved the study protocol (14-2019-H) as part of the project PSI2017-84170-R. Participants gave informed consent and were assessed in accordance with the Helsinki declaration.

Data accessibility. The datasets supporting this article have been uploaded as part of the electronic supplementary material.

Authors' contributions. A.M.R. substantially contributed to conception and design; analysis and interpretation of data; revising it critically for important intellectual content; final approval of the version to be published and agreement to be accountable for all aspects of the work in ensuring that questions related to the accuracy or integrity of any part of the work are appropriately investigated and resolved. M.J.G.C. substantially contributed to conception and design; acquisition and interpretation of data; drafting the article; final approval of the version to be published and agreement to be accountable for all aspects of the work in ensuring that questions related to the accuracy or

integrity of any part of the work are appropriately investigated and resolved. R.C. substantially contributed to conception, design and interpretation of data; revising it critically for important intellectual content; final approval of the version to be published and agreement to be accountable for all aspects of the work in ensuring that questions related to the accuracy or integrity of any part of the work are appropriately investigated and resolved. R.G.L. substantially contributed to conception, design and interpretation of data; revising it critically for important intellectual content; final approval of the version to be published and agreement to be accountable for all aspects of the work in ensuring that questions related to the accuracy or integrity of any part of the work are appropriately investigated and resolved. S.B.C. substantially contributed to the interpretation of data; revising it critically for important intellectual content; final approval of the version to be published and agreement to be accountable for all aspects of the work in ensuring that questions related to the accuracy or integrity of any part of the work are appropriately investigated and resolved. P.F.B. substantially contributed to conception, design and interpretation of data; revising it critically for important intellectual content; final approval of the version to be published and agreement to be accountable for all aspects of the work in ensuring that questions related to the accuracy or integrity of any part of the work are appropriately investigated and resolved.

Competing interests. We declare we have no competing interests.

Funding. This work was supported by The Spanish Ministry of Economy, Industry and Competitiveness (project: PSI2017-84170-R to P.F.B.), by Junta de Andalucía (project: UMA18-FEDERJA-137 to A.M.R and project: UMA18-FEDERJA-114 to P.F.B and R.C.), and by the Spanish Ministry of Education (FPU grant FPU15/05179 to R.G.L.). S.B.C. was funded by the Autism Research Trust, the Wellcome Trust, the Templeton World Charitable Foundation and the NIHR Biomedical Research Centre in Cambridge, during the period of this work. S.B.C. received funding from the Innovative Medicines Initiative 2 Joint Undertaking (JU) under grant agreement no. 777394. The J.U. receives support from the European Union's Horizon 2020 research and innovation programme and EFPIA and AUTISM SPEAKS, Autistica, SFARI. This study was funded by Innovation and Development Agency of Andalusia (grant no. SEJ-07325).

Acknowledgements. S.B.C.'s research was supported by the National Institute for Health Research (NIHR) Collaboration for Leadership in Applied Health Research and Care East of England at Cambridgeshire and Peterborough NHS Foundation Trust. The views expressed are those of the author(s) and not necessarily those of the NHS, NIHR or Department of Health and Social Care.

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
