## [Reviewer comments · Royal Society Open Science]

Review History

RSOS-191912.R0 (Original submission)

Review form: Reviewer 1

Is the manuscript scientifically sound in its present form?

No

Are the interpretations and conclusions justified by the results?

No

Is the language acceptable?

Yes

Do you have any ethical concerns with this paper?

No

Have you any concerns about statistical analyses in this paper?

No

Recommendation?

Major revision is needed (please make suggestions in comments)

Comments to the Author(s)

This manuscript investigates relations between two popular measures, the Reading the Mind in the Eye and the MSCEIT. The strengths of this study are its large sample size. I also appreciate that all data will be made open access.

I also have a few concerns.

In the introduction, I would appreciate a brief comparison of the RMET with other ToM measures, because they have quite different designs and approaches towards measuring ToM. A meta-analysis by Kirkland et al (2012) found only a weak relation between the RMET with other ToM tests. Thus, does RMET investigate a certain kind of ToM that is different from what is assessed by other ToM tests? (e.g., Strange Stories, Faux Pas, Movie for the Assessment of Social Cognition).

In the introduction, I would also appreciate a review of literature supporting the construct validity of the MSCEIT. There have been several critiques of the test (see work by Andrew Maul and Richard Roberts).

In the methods, please explain how the participants were recruited. Also, reliability coefficients describe the reliability of a test for a sample, but not the test itself. Thus, please report cronbach's alpha for the tests, including all subscales (e.g., easy RMET items) for the present sample.

Please list which items were in the easy and hard RMET subscales.

The investigation of relations between RMET and MSCEIT is pretty exploratory. I understand concrete hypotheses were probably not made beforehand. Thus, I would appreciate it if the discussion included references to studies that also looked at the relations between these two tests, as a way to evaluate the generalizability and replicability of the findings.

I strongly suggest the authors consider applying structural equation modeling to investigate relations between the tests, instead of the stepwise approach. That would allow them to model collinearity between the predictors (avoiding any issues of multicollinearity). It would also allow them to isolate unique relations (e.g., try a higher order EI factor indicated by the branches and see if RMET relates to only the higher order EI factor or also with the residual variance of the branches).

While the authors conclusively show that performance on the two tests is related, I don't think they can conclude that this means RMET measures the ability to understand mental states. As three meta-analyses have shown, performance on the MSCEIT is weakly to moderately correlated with general intelligence, including fluid and crystallized intelligence. Several studies have shown the RMET is also correlated with fluid and crystallized intelligence. Thus, it may be that fluid and crystallized intelligence act as a third variable and support performance on both tests. Thus, I wonder if intelligence is controlled for, what is the remaining relation between RMET and MSCEIT?

Finally, I am surprised that RMET isn't more highly correlated with the emotion perception branch of the MSCEIT, given some have used RMET as an emotion perception measure, or declared it as such (e.g., Oakley et al. 2016). However, as Fiori et al (2014, PLoS One) argued, because of the response options, participants get rewarded for not perceiving an emotion. Also, Olderbak et al. (2019, emotion review) found this branch was more weakly related with fluid and crystallized intelligence, compared with other emotion perception tests. Thus, the MSCEIT emotion perception test operates differently than other emotion perception ability tests.

In Table 1, this is Cohen's what? d ?

Finally, in evaluating a gender effect on performance on the RMET easy and hard items, was gender taken into account when identifying the easy and hard items? It may be that some items are easy for women, but hard for men, and vice versa.

Review form: Reviewer 2

Is the manuscript scientifically sound in its present form?

Yes

Are the interpretations and conclusions justified by the results?

Yes

Is the language acceptable?

Yes

Do you have any ethical concerns with this paper?

No

Have you any concerns about statistical analyses in this paper?

No

Recommendation?

Major revision is needed (please make suggestions in comments)

Comments to the Author(s)

This manuscript describes a study on the relationship between a Theory of Mind task and emotional intelligence (EI) in a sample of 874 Spanish speaking participants.

The topic of the manuscript is certainly of interest for both the ToM and EI literature. I believe the authors could better emphasize the most interesting and novel features of the study by fully integrating previous studies on the subject. Indeed, the question of how ToM is related to EI has already been investigated, and although the authors have acknowledged previous studies (e.g., Warrier et al., 2018) they have not integrated in the theoretical background of the study at least two key articles on this topic: Ferguson & Austin (2010) and Vesely Maillefer, Udayar & Fiori (2018).

The way the study is currently presented is more like a replication than a new addition to the literature. I encourage the authors to refine hypotheses by identifying how the current article adds to the existing literature.

For example, they could develop the rationale behind the choice to analyze predictors of easy and difficult items and expand the discussion regarding the results found for the two categories of items. I am wondering whether the authors could get even more conclusive results by selecting the very difficult and very easy items – such as the 12 most difficult and easiest. My consideration derives from the observation that the average score of the easy ($M = .81$) and difficult items ($M = .67$) does not seem very different.

An additional possibility is to expand the topic of gender differences in ToM and EI. I see that there was an imbalanced percentage of females and males, at the same time perhaps the sample is large enough to conduct additional analyses on this subject.

Please add a table with the full results of the hierarchical regressions.

In sum, I think the way the manuscript is currently framed is not as effective as it could be. Similar research questions the authors aim to answer with the study have been previously addressed. The authors could expand on the interesting features of the current study, such as the idea to compare predictors of the easiest and most difficult ToM items and refine the paper around this in order to make a more significant contribution to the literature.

Decision letter (RSOS-191912.R0)

09-Mar-2020

Dear Dr Gutiérrez Cobo,

The editors assigned to your paper ("The "Reading the mind in the Eyes" test and Emotional Intelligence") have now received comments from reviewers. We would like you to revise your paper in accordance with the referee and Associate Editor suggestions which can be found below (not including confidential reports to the Editor). Please note this decision does not guarantee eventual acceptance.

Please submit a copy of your revised paper before 01-Apr-2020. Please note that the revision deadline will expire at 00.00am on this date. If we do not hear from you within this time then it will be assumed that the paper has been withdrawn. In exceptional circumstances, extensions may be possible if agreed with the Editorial Office in advance. We do not allow multiple rounds of revision so we urge you to make every effort to fully address all of the comments at this stage. If deemed necessary by the Editors, your manuscript will be sent back to one or more of the original reviewers for assessment. If the original reviewers are not available, we may invite new reviewers.

- Data accessibility

It is a condition of publication that all supporting data are made available either as supplementary information or preferably in a suitable permanent repository. The data accessibility section should state where the article's supporting data can be accessed. This section should also include details, where possible of where to access other relevant research materials

such as statistical tools, protocols, software etc can be accessed. If the data have been deposited in an external repository this section should list the database, accession number and link to the DOI for all data from the article that have been made publicly available. Data sets that have been deposited in an external repository and have a DOI should also be appropriately cited in the manuscript and included in the reference list.

If you wish to submit your supporting data or code to Dryad (<http://datadryad.org/>), or modify your current submission to dryad, please use the following link:
<http://datadryad.org/submit?journalID=RSOS&manu=RSOS-191912>

- **Competing interests**

- **Authors' contributions**

- **Acknowledgements**

- **Funding statement**

Kind regards,

Lianne Parkhouse

Editorial Coordinator

on behalf of Dr Antonia Hamilton (Associate Editor) and Essi Viding (Subject Editor)

Reviewers' Comments to Author:

Reviewer: 1

Comments to the Author(s)

This manuscript investigates relations between two popular measures, the Reading the Mind in

the Eye and the MSCEIT. The strengths of this study are its large sample size. I also appreciate that all data will be made open access.

I also have a few concerns.

In the introduction, I would appreciate a brief comparison of the RMET with other ToM measures, because they have quite different designs and approaches towards measuring ToM. A meta-analysis by Kirkland et al (2012) found only a weak relation between the RMET with other ToM tests. Thus, does RMET investigate a certain kind of ToM that is different from what is assessed by other ToM tests? (e.g., Strange Stories, Faux Pas, Movie for the Assessment of Social Cognition).

In the introduction, I would also appreciate a review of literature supporting the construct validity of the MSCEIT. There have been several critiques of the test (see work by Andrew Maul and Richard Roberts).

In the methods, please explain how the participants were recruited. Also, reliability coefficients describe the reliability of a test for a sample, but not the test itself. Thus, please report cronbach's alpha for the tests, including all subscales (e.g., easy RMET items) for the present sample.

Please list which items were in the easy and hard RMET subscales.

The investigation of relations between RMET and MSCEIT is pretty exploratory. I understand concrete hypotheses were probably not made beforehand. Thus, I would appreciate it if the discussion included references to studies that also looked at the relations between these two tests, as a way to evaluate the generalizability and replicability of the findings.

I strongly suggest the authors consider applying structural equation modeling to investigate relations between the tests, instead of the stepwise approach. That would allow them to model collinearity between the predictors (avoiding any issues of multicollinearity). It would also allow them to isolate unique relations (e.g., try a higher order EI factor indicated by the branches and see if RMET relates to only the higher order EI factor or also with the residual variance of the branches).

While the authors conclusively show that performance on the two tests is related, I don't think they can conclude that this means RMET measures the ability to understand mental states. As three meta-analyses have shown, performance on the MSCEIT is weakly to moderately correlated with general intelligence, including fluid and crystallized intelligence. Several studies have shown the RMET is also correlated with fluid and crystallized intelligence. Thus, it may be that fluid and crystallized intelligence act as a third variable and support performance on both tests. Thus, I wonder if intelligence is controlled for, what is the remaining relation between RMET and MSCEIT?

Finally, I am surprised that RMET isn't more highly correlated with the emotion perception branch of the MSCEIT, given some have used RMET as an emotion perception measure, or declared it as such (e.g., Oakley et al. 2016). However, as Fiori et al (2014, PLoS one) argued, because of the response options, participants get rewarded for not perceiving an emotion. Also, Olderbak et al. (2019, emotion review) found this branch was more weakly related with fluid and crystallized intelligence, compared with other emotion perception tests. Thus, the MSCEIT emotion perception test operates differently than other emotion perception ability tests.

In Table 1, this is Cohen's d ?

Finally, in evaluating a gender effect on performance on the RMET easy and hard items, was gender taken into account when identifying the easy and hard items? It may be that some items are easy for women, but hard for men, and vice versa.

Reviewer: 2

Comments to the Author(s)

This manuscript describes a study on the relationship between a Theory of Mind task and emotional intelligence (EI) in a sample of 874 Spanish speaking participants.

The topic of the manuscript is certainly of interest for both the ToM and EI literature. I believe the authors could better emphasize the most interesting and novel features of the study by fully integrating previous studies on the subject. Indeed, the question of how ToM is related to EI has already been investigated, and although the authors have acknowledged previous studies (e.g., Warrier et al., 2018) they have not integrated in the theoretical background of the study at least two key articles on this topic: Ferguson & Austin (2010) and Vesely Maillefer, Udayar & Fiori (2018).

The way the study is currently presented is more like a replication than a new addition to the literature. I encourage the authors to refine hypotheses by identifying how the current article adds to the existing literature.

For example, they could develop the rationale behind the choice to analyze predictors of easy and difficult items and expand the discussion regarding the results found for the two categories of items. I am wondering whether the authors could get even more conclusive results by selecting the very difficult and very easy items – such as the 12 most difficult and easiest. My consideration derives from the observation that the average score of the easy ($M = .81$) and difficult items ($M = .67$) does not seem very different.

An additional possibility is to expand the topic of gender differences in ToM and EI. I see that there was an imbalanced percentage of females and males, at the same time perhaps the sample is large enough to conduct additional analyses on this subject.

Please add a table with the full results of the hierarchical regressions.

In sum, I think the way the manuscript is currently framed is not as effective as it could be. Similar research questions the authors aim to answer with the study have been previously addressed. The authors could expand on the interesting features of the current study, such as the idea to compare predictors of the easiest and most difficult ToM items and refine the paper around this in order to make a more significant contribution to the literature.

Author's Response to Decision Letter for (RSOS-191912.R0)

See Appendix A.

RSOS-191912.R1 (Revision)

Review form: Reviewer 3

Is the manuscript scientifically sound in its present form?

No

Are the interpretations and conclusions justified by the results?

No

Is the language acceptable?

Yes

Do you have any ethical concerns with this paper?

No

Have you any concerns about statistical analyses in this paper?

Yes

Recommendation?

Major revision is needed (please make suggestions in comments)

Comments to the Author(s)

In the current manuscript the authors describe an investigation of the relationship between the Reading the Mind in the Eye test (Eyes Test) and the MSCEIT.

I have a few concerns.

1. My first major concern is about the interpretation of the current results. The conclusion that the Eyes Test is as useful as a ToM measure based on the current analysis cannot be made (already mentioned by Reviewer 1 comment 8). The understanding test score only accounts for 8% of the variation in Eyes Test scores. Even including the other predictors, only 12% of the variation can be accounted for. The R squared is even smaller for the difficult items. These results raise a major question that is not discussed by the authors: what other individual differences account for the rest of the variance in the Eyes test performance? Only discussed in the introduction is the issue that the MSCEIT emotion perception task differs from other emotion perception tasks (see also my third comment about the MSCEIT). What does this mean for the interpretation of the results? Some studies (Olderbak et al., 2015; e.g., Peterson & Miller, 2012) find higher relations of the Eyes Test with vocabulary than with emotion perception. The results of the current study show the highest relation of the eyes test with the understanding task of the MSCEIT. The understanding task is described to measure "emotional vocabulary and understanding of how emotions may combine or change over time" (MHS.com/MSCEIT). A meta-analysis showed that the understanding emotions branch is the one that has the strongest relation with fluid and crystallized intelligence (Olderbak et al., 2019). Therefore, the correlation between the Eyes Test and the understanding task might be explained by the positive manifold of cognitive tasks and the lower correlation with the perception task in the current study might be explained by the lower correlation of the MSCEIT perception task with intelligence compared to other perception tasks. However, other studies also report low to no relation of the Eyes Test with emotion perception measures. Altogether, the conclusion that the current results validate the Eyes Test as a ToM measure is far-fetched. An integration of the Eyes Task in its nomological net including intelligence tasks as well as other ToM measures would be needed.

2. The statistical analysis: There are many issues with stepwise multiple regression analysis, which have been described in length by many authors (e.g., by Frank Harrell in Regression Modeling Strategies 2001). In response to reviewer 1 point 7 the authors report that they have performed a Structural Equation Model Analysis, but that there is no multicollinearity problem and it is easier to report the results of the stepwise regression. However, multicollinearity is not the only issue with stepwise regression. Furthermore, reporting the results within one latent structural equation model seems easier than reporting multiple models for the stepwise regression analysis. In addition, structural equation modelling has many advantages like taking measurement error into account. The sample size actually allows for such an analysis. Due to the problems with stepwise methods and the advantages of latent variable modelling, I would highly recommend using the latter.

3. MSCEIT: the examples of problems with the MCSEIT described in the introduction are by far not the most pronounced. The MSCEIT uses consensus rating instead of veridical answers, thereby measuring conformity with majority judgements. A second major issue is the opaque

estimation of scores. Furthermore, it is questionable if the calculation of a total score for MSCEIT is justifiable. Some of the correlations between branches in the current study were quite low (e.g., between perception and understanding). Confirmatory factor analysis has shown that correlations between latent factors for the four branches are low, and a higher order model does not fit the data well (e.g., Fiori & Antonakis, 2011). These results question the assumption that a global EI factor can be extracted from the MSCEIT. Associated, in the current version only the reliability for the total score is reported, although later on the subtest scores are used for the main analysis. The reliabilities for the subtest scores should be reported as well.

4. Reliability of the Eyes Test: Cronbach's alpha can be estimated for dichotomous items and a variation in difficulty is also not problematic (answer to Reviewer 1's fourth comment). The Eyes Test is described to measure one construct (Baron-Cohen et al., 2001). Therefore, the inter-item correlations should reflect this common construct. The correlation between the easy and difficult items in Table 2 is pretty low. Olderbak et al. (2015) discuss the issue of low internal consistency of the Eyes test and show that it is possible to create a version of the test that is more homogeneous. This is also related to the first issue about the conclusion of the results. On that note, it would be interesting to learn more about the items that are in the easy and difficult subscales. How are the emotions distributed in the scales? How about other characteristics of the stimuli like gender, lighting etc.? Like already mentioned by Reviewer 2 (comment 2), the mean of the hard and easy scales do not really differ. The distribution from both scale scores are left skewed and strongly overlap. More information about how these scales were put together would be necessary. It is unclear why the authors decided to use the results of a previous study for the item categorization rather than the current data. Furthermore, it should be mentioned that using dichotomization always leads to loss of information.

5. Replicability: It is not enough to report that moderation effects of gender were not significant. Estimates of the models should at least be reported in the Appendix. To make the analysis replicable the SPSS syntax for all models could be made open access.

In summary, I think that the manuscript needs major revisions to meet the criteria of the Royal Society Open Science for publication.

Baron-Cohen, S., Wheelwright, S., Hill, J., Raste, Y., & Plumb, I. (2001). The "Reading the Mind in the Eyes" Test Revised Version: A Study with Normal Adults, and Adults with Asperger Syndrome or High-functioning Autism. *Journal of Child Psychology and Psychiatry*, 42(2), 241-251. <https://doi.org/10.1111/1469-7610.00715>

Fiori, M., & Antonakis, J. (2011). The ability model of emotional intelligence: Searching for valid measures. *Personality and Individual Differences*, 50(3), 329-334. <https://doi.org/10.1016/j.paid.2010.10.010>

Olderbak, S., Semmler, M., & Doebler, P. (2019). Four-branch model of ability emotional intelligence with fluid and crystallized intelligence: A meta-analysis of relations. *Emotion Review*, 11(2), 166-183.

Olderbak, S., Wilhelm, O., Oлару, G., Geiger, M., Brenneman, M. W., & Roberts, R. D. (2015). A psychometric analysis of the reading the mind in the eyes test: Toward a brief form for research and applied settings. *Frontiers in Psychology*, 6, 1503.

Peterson, E., & Miller, S. (2012). The Eyes Test as a Measure of Individual Differences: How much of the Variance Reflects Verbal IQ? *Frontiers in Psychology*, 3. <https://doi.org/10.3389/fpsyg.2012.00220>

Decision letter (RSOS-191912.R1)

Dear Dr Gutiérrez Cobo:

Manuscript ID RSOS-191912.R1 entitled "The "Reading the mind in the Eyes" test and Emotional Intelligence" which you submitted to Royal Society Open Science, has been reviewed. The comments from reviewer(s) are included at the bottom of this letter.

In view of the criticisms of the reviewer(s), I must decline the manuscript for publication in Royal Society Open Science at this time. However, a new manuscript may be submitted which takes into consideration these comments.

Please note that resubmitting your manuscript does not guarantee eventual acceptance, and that your resubmission will be subject to re-review by the reviewer(s) before a decision is rendered.

You will be unable to make your revisions on the originally submitted version of your manuscript. Instead, revise your manuscript using a word processing program and save it on your computer.

You may also click the below link to start the resubmission process (or continue the process if you have already started your resubmission) for your manuscript. If you use the below link you will not be required to login to ScholarOne Manuscripts.

*** PLEASE NOTE: This is a two-step process. After clicking on the link, you will be directed to a webpage to confirm. ***

https://mc.manuscriptcentral.com/rsos?URL_MASK=fa54404587d746528821331cc21d6787

Because we are trying to facilitate timely publication of manuscripts submitted to Royal Society Open Science, your resubmitted manuscript should be submitted by 23-Dec-2020. If you are unable to submit by this date please contact the Editorial Office for options.

I look forward to a resubmission.

on behalf of Prof Essi Viding (Subject Editor)
openscience@royalsociety.org

Associate Editor Comments to Author:

We've received one referee report on your revised paper which raises several concerns about your manuscript. Please ensure that you address all these concerns in a detailed point-by-point response upon resubmission.

Reviewer comments to Author:

Reviewer: 3

Comments to the Author(s)

In the current manuscript the authors describe an investigation of the relationship between the Reading the Mind in the Eye test (Eyes Test) and the MSCEIT.

I have a few concerns.

1. My first major concern is about the interpretation of the current results. The conclusion that the Eyes Test is as useful as a ToM measure based on the current analysis cannot be made (already mentioned by Reviewer 1 comment 8). The understanding test score only accounts for 8% of the variation in Eyes Test scores. Even including the other predictors, only 12% of the variation can be accounted for. The R squared is even smaller for the difficult items. These results raise a major question that is not discussed by the authors: what other individual differences account for the rest of the variance in the Eyes test performance? Only discussed in the introduction is the issue that the MSCEIT emotion perception task differs from other emotion perception tasks (see also my third comment about the MSCEIT). What does this mean for the interpretation of the results? Some studies (Olderbak et al., 2015; e.g., Peterson & Miller, 2012) find higher relations of the Eyes Test with vocabulary than with emotion perception. The results of the current study show the highest relation of the eyes test with the understanding task of the MSCEIT. The understanding task is described to measure “emotional vocabulary and understanding of how emotions may combine or change over time” (MHS.com/MSCEIT). A meta-analysis showed that the understanding emotions branch is the one that has the strongest relation with fluid and crystallized intelligence (Olderbak et al., 2019). Therefore, the correlation between the Eyes Test and the understanding task might be explained by the positive manifold of cognitive tasks and the lower correlation with the perception task in the current study might be explained by the lower correlation of the MSCEIT perception task with intelligence compared to other perception tasks. However, other studies also report low to no relation of the Eyes Test with emotion perception measures. Altogether, the conclusion that the current results validate the Eyes Test as a ToM measure is far-fetched. An integration of the Eyes Task in its nomological net including intelligence tasks as well as other ToM measures would be needed.
2. The statistical analysis: There are many issues with stepwise multiple regression analysis, which have been described in length by many authors (e.g., by Frank Harrell in *Regression Modeling Strategies* 2001). In response to reviewer 1 point 7 the authors report that they have performed a Structural Equation Model Analysis, but that there is no multicollinearity problem and it is easier to report the results of the stepwise regression. However, multicollinearity is not the only issue with stepwise regression. Furthermore, reporting the results within one latent structural equation model seems easier than reporting multiple models for the stepwise regression analysis. In addition, structural equation modelling has many advantages like taking measurement error into account. The sample size actually allows for such an analysis. Due to the problems with stepwise methods and the advantages of latent variable modelling, I would highly recommend using the latter.
3. MSCEIT: the examples of problems with the MSCEIT described in the introduction are by far not the most pronounced. The MSCEIT uses consensus rating instead of veridical answers, thereby measuring conformity with majority judgements. A second major issue is the opaque estimation of scores. Furthermore, it is questionable if the calculation of a total score for MSCEIT is justifiable. Some of the correlations between branches in the current study were quite low (e.g., between perception and understanding). Confirmatory factor analysis has shown that correlations between latent factors for the four branches are low, and a higher order model does not fit the data well (e.g., Fiori & Antonakis, 2011). These results question the assumption that a global EI factor can be extracted from the MSCEIT. Associated, in the current version only the reliability for the total score is reported, although later on the subtest scores are used for the main analysis. The reliabilities for the subtest scores should be reported as well.
4. Reliability of the Eyes Test: Cronbach’s alpha can be estimated for dichotomous items and a variation in difficulty is also not problematic (answer to Reviewer 1’s fourth comment). The Eyes Test is described to measure one construct (Baron-Cohen et al., 2001). Therefore, the inter-item correlations should reflect this common construct. The correlation between the easy and difficult items in Table 2 is pretty low. Olderbak et al. (2015) discuss the issue of low internal consistency of the Eyes test and show that it is possible to create a version of the test that is more homogeneous. This is also related to the first issue about the conclusion of the results. On that note, it would be interesting to learn more about the items that are in the easy and difficult subscales. How are the emotions distributed in the scales? How about other characteristics of the stimuli like gender, lighting etc.? Like already mentioned by Reviewer 2 (comment 2), the mean

of the hard and easy scales do not really differ. The distribution from both scale scores are left skewed and strongly overlap. More information about how these scales were put together would be necessary. It is unclear why the authors decided to use the results of a previous study for the item categorization rather than the current data. Furthermore, it should be mentioned that using dichotomization always leads to loss of information.

5. Replicability: It is not enough to report that moderation effects of gender were not significant. Estimates of the models should at least be reported in the Appendix. To make the analysis replicable the SPSS syntax for all models could be made open access.

In summary, I think that the manuscript needs major revisions to meet the criteria of the Royal Society Open Science for publication.

Baron-Cohen, S., Wheelwright, S., Hill, J., Raste, Y., & Plumb, I. (2001). The "Reading the Mind in the Eyes" Test Revised Version: A Study with Normal Adults, and Adults with Asperger Syndrome or High-functioning Autism. *Journal of Child Psychology and Psychiatry*, 42(2), 241–251. <https://doi.org/10.1111/1469-7610.00715>

Fiori, M., & Antonakis, J. (2011). The ability model of emotional intelligence: Searching for valid measures. *Personality and Individual Differences*, 50(3), 329–334.

<https://doi.org/10.1016/j.paid.2010.10.010>

Olderbak, S., Semmler, M., & Doebler, P. (2019). Four-branch model of ability emotional intelligence with fluid and crystallized intelligence: A meta-analysis of relations. *Emotion Review*, 11(2), 166–183.

Olderbak, S., Wilhelm, O., Olaru, G., Geiger, M., Brenneman, M. W., & Roberts, R. D. (2015). A psychometric analysis of the reading the mind in the eyes test: Toward a brief form for research and applied settings. *Frontiers in Psychology*, 6, 1503.

Peterson, E., & Miller, S. (2012). The Eyes Test as a Measure of Individual Differences: How much of the Variance Reflects Verbal IQ? *Frontiers in Psychology*, 3.

<https://doi.org/10.3389/fpsyg.2012.00220>

Author's Response to Decision Letter for (RSOS-191912.R1)

See Appendix B.

RSOS-201305.R0

Review form: Reviewer 3

Is the manuscript scientifically sound in its present form?

Yes

Are the interpretations and conclusions justified by the results?

Yes

Is the language acceptable?

Yes

Do you have any ethical concerns with this paper?

No

Have you any concerns about statistical analyses in this paper?

No

Recommendation?

Accept as is

Comments to the Author(s)

In their revision the authors included most of my suggestions. I still think that the conclusion could be even more cautionary. However, they report all the necessary information for every reader to come to their own conclusions.

Decision letter (RSOS-201305.R0)

Dear Dr Gutiérrez Cobo,

I am pleased to inform you that your manuscript entitled "The "Reading the mind in the Eyes" test and Emotional Intelligence" is now accepted for publication in Royal Society Open Science.

Royal Society Open Science operates under a continuous publication model. Your article will be published as soon as it is ready for publication, and this will be the final version of the paper. As such, it can be cited immediately by other researchers. As the issue version of your paper will be the only version to be published I would advise you to check your proofs thoroughly as changes cannot be made once the paper is published.

Articles are normally press released. For this to be effective we set an embargo on news coverage corresponding to the publication date of the article. We request that news media and the authors do not publish stories ahead of this embargo (when final version of the article is available). Please see the Royal Society Publishing guidance on how you may share your accepted author manuscript at <https://royalsociety.org/journals/ethics-policies/media-embargo/>.

on behalf of the Associate Editor and Professor Essi Viding (Subject Editor)

Reviewer comments to Author:

Reviewer: 3

Comments to the Author(s)

In their revision the authors included most of my suggestions. I still think that the conclusion could be even more cautionary. However, they report all the necessary information for every reader to come to their own conclusions.

Appendix A

Manuscript Number: RSOS-191912

Dear Editors

With regard to the manuscript entitled “The “Reading the mind in the Eyes” test and Emotional Intelligence” we would like to thank you and the reviewers for your valuable comments and suggestions. We think that these have helped us to improve the quality of our manuscript. We hope that we have been able to adequately address the questions raised by the reviewers and that the manuscript is now suitable for publication in Royal Society Open Science. Changes/additions to the paper have been marked **in red**.

Comments from the referees

Answers to the referees' comments in blue.

Referees Comments to Author:

Reviewer 1

1. This manuscript investigates relations between two popular measures, the Reading the Mind in the Eye and the MSCEIT. The strengths of this study are its large sample size. I also appreciate that all data will be made open access.

- Thank you for the careful reading of the manuscript, it is much appreciated

I also have a few concerns.

2. In the introduction, I would appreciate a brief comparison of the RMET with other ToM measures, because they have quite different designs and approaches towards measuring ToM. A meta-analysis by Kirkland et al (2012) found only a weak relation between the RMET with other ToM tests. Thus, does RMET investigate a certain kind of ToM that is different from what is assessed by other ToM tests? (e.g., Strange Stories, Faux Pas, Movie for the Assessment of Social Cognition).

- We have now included a brief explanation of some of the different approaches used for evaluating ToM in the Introduction section (page 2, paragraph 1).

3. In the introduction, I would also appreciate a review of literature supporting the construct validity of the MSCEIT. There have been several critiques of the test (see work by Andrew Maul and Richard Roberts).

- We have added a brief review including the advantages and limitations of the MSCEIT (page 2, paragraph 2).

4. In the methods, please explain how the participants were recruited. Also, reliability coefficients describe the reliability of a test for a sample, but not the test itself. Thus, please report cronbach's alpha for the tests, including all subscales (e.g., easy RMET items) for the present sample.

- Thank you for your suggestions. We have now described how the participants were recruited.

With respect to the reliability of the Eyes Test instrument, previous literature has shown that Cronbach's alpha is not an appropriate measure of reliability for this instrument due to the particular characteristics of the scale (i.e. it is dichotomous and there are significant variations in difficulty between the items). Thus, the reliability of this instrument has usually been assessed by test-retest procedures. However, given the cross-sectional nature

of the current study, it was not possible to compute test-retest reliability for our sample. However, the version of the instrument that we used in this research has been shown to have acceptable test-retest reliability in a Spanish sample with similar socio-demographic characteristics to the one studied here (intra-class correlation coefficient = 0.63; see Fernández-Abascal et al., 2013).

Finally, the internal consistency of the MSCEIT instrument has now been included.

Fernández-Abascal EG, Cabello R, Fernández-Berrocal P, Baron-cohen S. Test-retest reliability of the ' Reading the Mind in the Eyes ' test : a one-year follow-up study. *Mol Autism*. 2013;4:33.

5. Please list which items were in the easy and hard RMET subscales.

- We have now added a description of the items included in the easy and difficult RMET subscales.

6. The investigation of relations between RMET and MSCEIT is pretty exploratory. I understand concrete hypotheses were probably not made beforehand. Thus, I would appreciate it if the discussion included references to studies that also looked at the relations between these two tests, as a way to evaluate the generalizability and replicability of the findings.

- We have included references to studies that analyze the relationships between EI and the Eyes Test in the Introduction section (page 3, paragraph 3).

7. I strongly suggest the authors consider applying structural equation modeling to investigate relations between the tests, instead of the stepwise approach. That would allow them to model collinearity between the predictors (avoiding any issues of multicollinearity). It would also allow them to isolate unique relations (e.g., try a higher order EI factor indicated by the branches and see if RMET relates to only the higher order EI factor or also with the residual variance of the branches).

- Thank you for this valuable comment. We have verified that the multicollinearity assumption was met for all the predictors included in the regression models. This information has now been reported in the new version of the manuscript. Results applying structural equation modelling (through AMOS software) were similar to the current regression analysis. The best fitting models were the same for both types of analysis. Consequently, we think that the use of stepwise regression analysis including the standard MSCEIT branches as predictors can be a more appropriate approach given its greater simplicity and ease of interpretation.

8. While the authors conclusively show that performance on the two tests is related, I don't think they can conclude that this means RMET measures the ability to understand mental states. As three meta-analyses have shown, performance on the MSCEIT is weakly to moderately correlated with general intelligence, including fluid and crystallized intelligence. Several studies have shown the RMET is also correlated with fluid and crystallized intelligence. Thus, it may be that fluid and crystallized intelligence act as a third variable and support performance on both tests. Thus, I wonder if intelligence is controlled for, what is the remaining relation between RMET and MSCEIT?

- We have now included this idea for future investigations (page 6, paragraph 5).

9. Finally, I am surprised that RMET isn't more highly correlated with the emotion perception branch of the MSCEIT, given some have used RMET as an emotion perception measure, or declared it as such (e.g., Oakley et al. 2016). However, as Fiori et al (2014, PLoS ONE) argued, because of the response options, participants get rewarded for not perceiving an emotion. Also, Olderbak et al. (2019, emotion review) found this branch was more weakly related with fluid and crystallized intelligence, compared with other emotion perception tests. Thus, the MSCEIT emotion perception test operates differently than other emotion perception ability tests.

- We have now included this information as a future line of research in order to address this limitation (page 7, paragraph 1). Additional details have also been included in the Introduction section (page 2, paragraph 2).

10. In Table 1, this is Cohen's d ?

- This error has now been corrected. Thank you.

11. Finally, in evaluating a gender effect on performance on the RMET easy and hard items, was gender taken into account when identifying the easy and hard items? It may be that some items are easy for women, but hard for men, and vice versa.

- Thank you for your suggestion. Unfortunately, the study from which the items were selected (Fernández-Abascal et al. 2013) does not offer an analysis of the performance of the participants according to gender. In any case, additional analyses considering gender as a moderating factor have now been included following the suggestions of Reviewer 2. These new analyses provide information about gender differences in more depth.

Reviewer 2

This manuscript describes a study on the relationship between a Theory of Mind task and emotional intelligence (EI) in a sample of 874 Spanish speaking participants.

- Thank you very much for your comments. We appreciate that you took the time to review our manuscript.

1. The topic of the manuscript is certainly of interest for both the ToM and EI literature. I believe the authors could better emphasize the most interesting and novel features of the study by fully integrating previous studies on the subject. Indeed, the question of how ToM is related to EI has already been investigated, and although the authors have acknowledged previous studies (e.g., Warrier et al., 2018) they have not integrated in the theoretical background of the study at least two key articles on this topic: Ferguson & Austin (2010) and Vesely Maillefer, Udayar & Fiori (2018).

- We have now included these two articles in the Introduction section (page 3, paragraph 2) and we have also emphasize the most novel aspects of our investigation (page 3, paragraph 4).

2. The way the study is currently presented is more like a replication than a new addition to the literature. I encourage the authors to refine hypotheses by identifying how the current article adds to the existing literature. For example, they could develop the rationale behind the choice to analyze predictors of easy and difficult items and expand the discussion regarding the results found for the two categories of items. I am wondering whether the authors could get even more conclusive results by selecting the very difficult and very easy items—such as the 12 most difficult and easiest. My consideration derives from the observation that the average score of the easy ($M = .81$) and difficult items ($M = .67$) does not seem very different.

- We have re-formulated the aim of the study by emphasizing the important role of those analyses focused on the complexity of the items (page 3, paragraph 3).

We have conducted new analyses including the 12 most difficult and easiest items in the computation of the Eyes Test scores. The best fitting models were similar to our previous analysis except that in the new stepwise regression analysis the explained variance in the final models was lower in comparison with the original analyses. Consequently, we would prefer to maintain the previous analysis. Thank you for your suggestion.

3. An additional possibility is to expand the topic of gender differences in ToM and EI. I see that there was an imbalanced percentage of females and males, at the same time perhaps the sample is large enough to conduct additional analyses on this subject.

- We have added gender as a new issue in the manuscript by analyzing the moderating effect of this variable on the relationship between EI and Eyes Test scores (page 3, paragraph 3; page 5, paragraph 1; page 7, paragraph 5).

4. Please add a table with the full results of the hierarchical regressions.

- Thank you for your suggestion. Three new tables describing the full results of the stepwise regression analyses have now been added as supplementary material.

5. In sum, I think the way the manuscript is currently framed is not as effective as it could be. Similar research questions the authors aim to answer with the study have been previously addressed. The authors could expand on the interesting features of the current study, such as the idea to compare predictors of the easiest and most difficult ToM items and refine the paper around this in order to make a more significant contribution to the literature.

- We hope that we have been able to adequately address the questions raised.

Appendix B

Answers to the referees' comments in blue.

Reviewer: 3

Comments to the Author(s)

In the current manuscript the authors describe an investigation of the relationship between the Reading the Mind in the Eye test (Eyes Test) and the MSCEIT.

Thank you for all of your valuable comments. We have tried to respond to all of them.

I have a few concerns.

1. My first major concern is about the interpretation of the current results. The conclusion that the Eyes Test is as useful as a ToM measure based on the current analysis cannot be made (already mentioned by Reviewer 1 comment 8). The understanding test score only accounts for 8% of the variation in Eyes Test scores. Even including the other predictors, only 12% of the variation can be accounted for. The R squared is even smaller for the difficult items. These results raise a major question that is not discussed by the authors: what other individual differences account for the rest of the variance in the Eyes test performance? Only discussed in the introduction is the issue that the MSCEIT emotion perception task differs from other emotion perception tasks (see also my third comment about the MSCEIT). What does this mean for the interpretation of the results? Some studies (Olderbak et al., 2015; e.g., Peterson & Miller, 2012) find higher relations of the Eyes Test with vocabulary than with emotion perception.

The results of the current study show the highest relation of the eyes test with the understanding task of the MSCEIT. The understanding task is described to measure “emotional vocabulary and understanding of how emotions may combine or change over time” (MHS.com/MSCEIT). A meta-analysis showed that the understanding emotions branch is the one that has the strongest relation with fluid and crystallized intelligence

(Olderbak et al., 2019). Therefore, the correlation between the Eyes Test and the understanding task might be explained by the positive manifold of cognitive tasks and the lower correlation with the perception task in the current study might be explained by the lower correlation of the MSCEIT perception task with intelligence compared to other perception tasks. However, other studies also report low to no relation of the Eyes Test with emotion perception measures. Altogether, the conclusion that the current results validate the Eyes Test as a ToM measure is far-fetched. An integration of the Eyes Task in its nomological net including intelligence tasks as well as other ToM measures would be needed.

In the Discussion section we have already mentioned the limitation of the small variance explained by the MSCEIT branches, as well as suggestions for future lines of investigation (including intelligence measures) in order to test alternative explanations due to third variables. In addition, we have included the references proposed.

2. The statistical analysis: There are many issues with stepwise multiple regression analysis, which have been described in length by many authors (e.g., by Frank Harrell in Regression Modeling Strategies 2001). In response to reviewer 1 point 7 the authors report that they have performed a Structural Equation Model Analysis, but that there is no multicollinearity problem and it is easier to report the results of the stepwise regression. However, multicollinearity is not the only issue with stepwise regression. Furthermore, reporting the results within one latent structural equation model seems easier than reporting multiple models for the stepwise regression analysis. In addition, structural equation modelling has many advantages like taking measurement error into account. The sample size actually allows for such an analysis. Due to the problems with stepwise methods and the advantages of latent variable modelling, I would highly recommend using the latter.

Thank you for this suggestion. We understand the advantages of working with latent variable models, but we think that maintaining the ordinary measures of the MSCEIT (a global score and the four branches) instead of creating new latent variables have additional benefits. The MSCEIT is a questionnaire that is widely used and well-validated (it is not without certain problems, but these limitations have been described in the published articles); thus, using the standard EI branches allows us to interpret the results in a standard way and enables a comparison between our findings and those reported in the previous literature.

3. MSCEIT: the examples of problems with the MCSEIT described in the introduction are by far not the most pronounced. The MSCEIT uses consensus rating instead of veridical answers, thereby measuring conformity with majority judgements. A second major issue is the opaque estimation of scores. Furthermore, it is questionable if the calculation of a total score for MSCEIT is justifiable. Some of the correlations between branches in the current study were quite low (e.g., between perception and understanding). Confirmatory factor analysis has shown that correlations between latent factors for the four branches are low, and a higher order model does not fit the data well (e.g., Fiori & Antonakis, 2011). These results question the assumption that a global EI factor can be extracted from the MSCEIT. Associated, in the current version only the reliability for the total score is reported, although later on the subtest scores are used for the main analysis. The reliabilities for the subtest scores should be reported as well.

Effectively, Fiori and Antonakis (2011) found that a higher order model does not fit the data well. However, there is a vast amount of other studies supporting the original theoretical model (e.g., Brackett & Mayer, 2003; Legree et al, 2014; MacCann et al, 2014). More specifically for our purposes, the Spanish version of the MSCEIT, which we employ in our study, has generated results that are again in line with the higher order model (Cabello et al, 2016; Extremera, Fernández-Berrocal & Salovey, 2006).

Extremera, N., Fernández-Berrocal, P. and Salovey, P. (2006). Spanish version of the Mayer-Salovey-Caruso Emotional Intelligence Test (MSCEIT). Version 2.0: Reliabilities, age and gender differences. *Psicothema*, 18, 42-48:

Cabello, R, Sorrel, M.A., Fernández-Pinto, I., Extremera, N., Fernández-Berrocal, P. (2016). Age and gender differences in ability emotional intelligence in adults: A cross-sectional study. *Dev Psychol*, 52, 1486–92.

Brackett, M.A. & Mayer, J.D. (2003). Convergent, discriminant and incremental validity of competing measures of emotional intelligence. *Personality and Social Psychology Bulletin*, 29, 1147-1158.

Legree, P. J., Psootka, J., Robbins, J., Roberts, R. D., Putka, D. J., & Mul-lins, H. M. (2014). Profile similarity metrics as an alternate framework to score rating-based tests: MSCEIT reanalyses. *Intelligence*, 47, 159–174. doi:10.1016/j.intell.2014.09.005:

MacCann, C., Joseph, D. L., Newman, D. A., & Roberts, R. D. (2014). Emotional intelligence is a second-stratum factor of intelligence: Evidence from hierarchical and bifactor models. *Emotion, 14*, 358–374:

4. Reliability of the Eyes Test: Cronbach's alpha can be estimated for dichotomous items and a variation in difficulty is also not problematic (answer to Reviewer 1's fourth comment). The Eyes Test is described to measure one construct (Baron-Cohen et al., 2001). Therefore, the inter-item correlations should reflect this common construct. The correlation between the easy and difficult items in Table 2 is pretty low. Olderbak et al. (2015) discuss the issue of low internal consistency of the Eyes test and show that it is possible to create a version of the test that is more homogeneous. This is also related to the first issue about the conclusion of the results. On that note, it would be interesting to learn more about the items that are in the easy and difficult subscales. How are the emotions distributed in the scales? How about other characteristics of the stimuli like gender, lighting etc.? Like already mentioned by Reviewer 2 (comment 2), the mean of the hard and easy scales do not really differ. The distribution from both scale scores are left skewed and strongly overlap. More information about how these scales were put together would be necessary. It is unclear why the authors decided to use the results of a previous study for the item categorization rather than the current data. Furthermore, it should be mentioned that using dichotomization always leads to loss of information.

In the new version of the manuscript, Cronbach's alpha was computed as a measure of internal consistency for the Eyes Test.

We have now discussed the issue about the low internal consistency of the Eyes test in the paragraphs that describe the limitations of the study. Moreover, we propose the possibility of using the brief form of the questionnaire created by Olderbak et al. (2015) in further studies.

In response to the following question mentioned by Reviewer 2: "I am wondering whether the authors could get even more conclusive results by selecting the very difficult and very easy items—such as the 12 most difficult and easiest", we conducted the analyses including only the 12 most difficult and easiest items in the computation of the Eyes Test scores. However, although the best fitting models were similar to the previous analysis with

16 items per condition, the explained variance in the final models was lower. In addition, a comparison of the means between the 16 hardest items and the 16 easiest items showed clear significant differences ($t(873) = 27.74, p < 0.0001$, Cohen's $d = 0.94$; this information is included in the new version of the manuscript). Consequently, we preferred to maintain the previous analysis.

To categorize the easy and difficult items, we decided to use the results from the study validating the Spanish version of the Eyes test (Fernández-Abascal et al., 2013) in order to avoid double dipping biases in the analysis (circular analysis). This information is included in the new version of the manuscript.

5. Replicability: It is not enough to report that moderation effects of gender were not significant. Estimates of the models should at least be reported in the Appendix. To make the analysis replicable the SPSS syntax for all models could be made open access.

Results for the moderation effects have been included in the Appendix. In addition, SPSS syntax for all analyses has been uploaded as Supplementary material.

In summary, I think that the manuscript needs major revisions to meet the criteria of the Royal Society Open Science for publication.

Baron- Cohen, S., Wheelwright, S., Hill, J., Raste, Y., & Plumb, I. (2001). The “Reading the Mind in the Eyes” Test Revised Version: A Study with Normal Adults, and Adults with Asperger Syndrome or High-functioning Autism. *Journal of Child Psychology and Psychiatry*, 42(2), 241–251. <https://doi.org/10.1111/1469-7610.00715>

Fiori, M., & Antonakis, J. (2011). The ability model of emotional intelligence: Searching for valid measures. *Personality and Individual Differences*, 50(3), 329–334. <https://doi.org/10.1016/j.paid.2010.10.010>

Olderbak, S., Semmler, M., & Doebler, P. (2019). Four-branch model of ability emotional

intelligence with fluid and crystallized intelligence: A meta-analysis of relations. *Emotion Review*, 11(2), 166–183.

Olderbak, S., Wilhelm, O., Olaru, G., Geiger, M., Brenneman, M. W., & Roberts, R. D. (2015). A psychometric analysis of the reading the mind in the eyes test: Toward a brief form for research and applied settings. *Frontiers in Psychology*, 6, 1503.

Peterson, E., & Miller, S. (2012). The Eyes Test as a Measure of Individual Differences: How much of the Variance Reflects Verbal IQ? *Frontiers in Psychology*, 3. <https://doi.org/10.3389/fpsyg.2012.00220>